# Quantification of the Ecological Value of Railroad Development Areas Using Logistic Regression Analysis

**DOI:** 10.3390/ijerph182211764

**Published:** 2021-11-09

**Authors:** Min-Kyeong Kim, Duckshin Park, Dong Yeob Kim

**Affiliations:** 1Railroad Test & Certification Division, Korea Railroad Research Institute, Uiwang 16105, Korea; mkkim15@krri.re.kr; 2Transportation Environmental Research Department, Korea Railroad Research Institute, Uiwang 16105, Korea; dspark@krri.re.kr; 3School of Civil, Architectural Engineering & Landscape Architecture, Sungkyunkwan University, Seobo-ro, Suwon 2066, Korea

**Keywords:** Environmental Conservation Value Assessment Map (ECVAM), kappa analysis, logistic regression, quantitative evaluation, railway natural ecological environment

## Abstract

According to the national railway network construction plan, Investment in railways has increased due to the need for environmentally friendly transportation, and the rail network is expanding throughout South Korea. Railway projects should be evaluated using strategic environmental impact assessments. In the “Guidelines for the Construction of Environment-friendly Railways”, seven priority headings that must be considered for railway projects are described. This guide notes that qualitative evaluation must be conducted during the survey process to reasonably predict impacts on the environment. However, quantitative evaluation with specific indicator values may also be necessary. In this study, independence analysis and logistic regression analysis were used to quantitatively evaluate railway environmental and ecological indicators. The results were used to develop a regression model reflecting seven indicators; biodiversity class, ecosystem type, vegetation conservation class, tree age class, ecological naturalness, presence of river ecosystems, and fragmented patch size. The fitness regression model showed 90.3% classification accuracy and the receiver operating curve (ROC) model fit was 88.6%. An environmental quality assessment map was prepared by classifying areas of environmental quality according to five grades. This is the first model for environmental and ecological evaluation of railway projects. Evaluation using the map showed that the railroad passes through areas with lower protection values compared to the results obtained using the national environmental evaluation map. Kappa analysis showed a low level of agreement between the two maps (kappa coefficient = 0.212). The results of this study can be applied to railway development project sites and may help to identify the best sites for the development of an environmentally friendly railway system.

## 1. Introduction

In South Korea, railway construction projects are subject to environmental impact assessments, which have six areas of evaluation: atmospheric environment, water environment, land environment, natural ecological environment, living environment, and socioeconomic environment [1]. A total of 21 headings ensure detailed evaluation. Among them, seven headings represent key evaluation items: air quality, water quality, topography/geology, fauna/flora, natural environmental assets, noise/vibration, and amusement/scenery. Some headings are excluded depending on the situation of a particular railroad route. Environmental standards are provided for air quality, water quality, and the aquatic ecosystem based on the Enforcement Decree of the Framework Act on Environmental Policy. Air quality standards are provided based on average values. Water quality standards are listed separately for river, lake, groundwater, and seawater, with health protection standards and living environment standards provided for each field. Article 63-2 of the Enforcement Rule of the Noise and Vibration Control Act provides noise standards (in dB) for each vehicle type when stopped and running [1,2].

However, for fauna/flora and natural environment assets important for environmental quality, field survey results are described qualitatively, and no specific values are listed. In addition, the environmental impact assessment of previous railway construction projects focused mainly on the living environment. Quantitative evaluation of the environmental quality of railway areas has rarely been conducted [3,4].

Descriptions of other aspects of environmental quality in the “Guidelines for the Construction of Environment-friendly Railways” are similar to those for fauna/flora and natural environment assets, which are considered important for the evaluation of environmentally friendly railways. The use of detours and mitigation measures has been suggested when the vegetation conservation grade is ≥2, the ecological nature grade is 1, and the railway passes through legally protected areas. Currently, some railways run through protected coastal areas (tidal flats) and environmental conservation areas, but such sites have not yet been reviewed. Consideration of domestic environmental resources and ecosystem types remains insufficient [1,3,4].

Current investigations of environmental impacts depend on qualitative evaluation and continuous data collection. When a railway project is investigated, evaluations using only qualitative information may have low objectivity, consistency, and efficiency. The relative diversity of spaces cannot be identified effectively from qualitative information, so quantitative evaluation is essential to evaluation. In particular, logistic regression analysis has been widely used to evaluate various environmental quality indicators that lack specific evaluation guidelines [5,6]. For effective environmental impact assessment, the development of sound indicators and quantitative evaluation measures that can be used to explain the dynamics of the project area is essential. Such measures will elucidate the impacts of railway projects using information obtained in field surveys. Kim et al. (2019) [4] suggested specific indicators and headings for evaluating the ecological aspects of railway projects. The present study used those indicators and headings to propose a method for developing environmental quality assessment maps through logistic regression analysis. The model used in this study aims to evaluate the environmental ecology of the railroad development project in comparison with the national land environment evaluation map and Kim et al. (2017) [5]. As for the railroad project, fragmented patch size, an item not included in other cases, was added as an ecosystem cut-off project, and through this, the ecosystem cut-off and fragmentation were evaluated. Current railway areas were investigated quantitatively using environmental quality values. The results can be applied to strategic environmental impact assessment for comparison and evaluation of alternative railway lines. Shortcomings in the environmental aspects of railway development can also be examined in advance with this method.

Recently, preliminary feasibility investigations have not been performed for domestic railway projects. The objective of this study is to develop quantitative measures for assessing candidate railway routes through evaluation of their environmental impacts within the framework of a strategic environmental impact assessment that examines the propriety of railway plans and the feasibility of their locations.

In this study, the existing regulations for evaluating the environmental ecology of the railway development site are not clear because the natural ecological environment is a factor that cannot be restored when damaged. To evaluate the environmental and ecological suitability of railway lines by developing environmental and ecological evaluation criteria and evaluation methods to select environmentally-friendly railway lines. And an environmental quality assessment map was prepared by classifying areas of environmental quality according to five grades. This is the first model for environmental and ecological evaluation of railway projects.

## 2. Methods

### 2.1. Literature Review

As a result of examining the research related to the environmental evaluation of railway projects related to the subject of this study, only the relative values of the evaluation items using the AHP method were presented, and there were not many related prior studies. Also, overseas, there are studies on the development of a methodology to map railway lines and surrounding land use using Unmanned Aerial Vehicle (UAVs) [6,7], but there are not many related prior studies.

Research on the environment of railways has generally focused on living conditions; little attention has been paid to environmental and ecological considerations [8,9,10,11]. Although Korea Environment Institute (KEI) [8] presented ecological items, it mainly conducted research on living conditions, and the contents, including the ecosystem, were presented only in the abstract, and thus the environmental and ecological aspects were not specifically reviewed. Kim [9] drew items for the overall railroad environmental impact assessment. Ser and Koo [10] presented indicators for evaluating railroad route targets only in terms of landscape ecology and did not suggest the process of deriving evaluation items and the relationship between evaluation items for each indicator. Lee et al. [11] presented evaluation items for the overall railway project using the analytical hierarchy process (AHP) method and suggested that the natural ecological environment field is an important item to review because of its high weight.

Related studies related to logistic regression analysis used in this study were reviewed. This method has been used to develop models based on geographic information systems (GIS) techniques, and to draw evaluation maps with various themes. It has also been used for quantitative evaluation in large or inaccessible areas. Logistic regression analysis has been used to evaluate ecological factors related to various topics [6,12,13,14,15,16,17]. In related case studies, logistic regression analysis was used in various fields, such as landslide risk assessment prediction [12,14,16], wild boar habitat model development [13], amphibians habitat suitability model [6], and forest impact assessment in North Korea [5,15]. However, as the subject of this study, there are no examples of use in the environmental evaluation of railway projects.

Looking at overseas research cases, there are few cases related to railway projects and environmental evaluation, and recently, there are examples of railroad lines using UAVs and land use around them [7]. It was argued that map development for the railway project was necessary, and Red-Green-Blue (RGB) and Normalized Difference Vegetation Index (NDVI) were analyzed to understand the current status of land use around railroad lines. Through this, it is suggested that it can be utilized for sustainable planning of urban environment and railroad operators.

### 2.2. Study Site

Gyeong-ui Central Line was selected as the study site, as it passes through both urban and non-urban areas, including areas of various ecotypes, and could provide a rail connection between North Korea and South Korea (Figure 1). The results of this study may provide a foundation for screening alternative railway lines at the early development stage of strategic environmental impact assessments.

### 2.3. Evaluation of the Environmental Quality Value

A total of five indicators, and 16 headings thereunder, have been suggested for evaluation of the environmental quality of railway project areas [3,4]. The 16 headings include species diversity grade, species richness, ecosystem type, areal distribution of vegetation conservation grade, tree age grade, ecological naturalness grade, presence of adjacent wetland ecosystems, presence of adjacent river ecosystems, size of fragmented patches, rate of patch split, the ratio of structures affected by fragmentation, number of gaps, presence of legally protected areas, number of adjacent protected areas, presence of endangered species habitat, and presence of endangered species.

To develop measures for evaluating those indicators and headings, spatial information was collected, and information maps were constructed for independence analysis and logistic regression analysis. Korean railway and road network information, provided by the National Transportation Database, were used as basic data. A vegetation map (1:25,000), ecological naturalness map (plant and animal distribution map), map of forest type (1:25,000), and soil map (1:25,000) were also used. Species abundance and habitat fragmentation, the presence and number of legally protected areas, and the presence of endangered species were assessed in raster form. Additional spatial data in vector form were also collected.

Spatial information was obtained for each evaluation factor, and data for the dependent and independent variables were prepared in a 30-m-resolution raster format. The most suitable resolution for land analysis is 20–30 m. ArcGIS 10.2(ESRI, Seoul, Korea) was used for the analysis. The study area was within 1 km of the Gyeong-ui Central Line [3,17]. Due to the linear form of the railway project, the raster-based point sampling method described by Kim et al. (2017) [17,18,19] was used. Valuable area for environmental quality conservation was the dependent variable in this study. Spatial information was converted into raster data of 30-m resolution for use as independent variables, and then into sampling points (Table 1). The independent variables, the grades suggested in each of the existing thematic maps were used for species diversity, vegetation conservation class, tree age class, and ecological naturalness class.

The attribute values were converted into points for analysis. The total number of sampling points was 274,689, with 37,729 points in environmentally favorable areas and 236,960 in unfavorable areas. Information on the presence of favorable areas and the attribute values of each independent variable was obtained. The dependent variables were determined based on the presence of environmentally favorable areas and the influence of the independent variables. The criteria for identifying environmentally favorable and unfavorable areas were grade 1 in the ecological naturalness class, grade ≥ 2 in the vegetation conservation class, and the presence of legally protected areas, as suggested in the “Guidelines for the Construction of Environment-friendly Railways”. If one of those criteria was satisfied, the area was considered an environmentally favorable area (Figure 2).

### 2.4. Independence Analysis

In general, strong correlations between independent variables result in low statistical significance due to multicollinearity. To increase the accuracy of analysis, testing for relationships among independent variables is necessary. Because the variables used in this study followed isometric and ratio scales, they were analyzed using Pearson’s correlation coefficient. The sampling values of 274,689 sampling points were analyzed using Pearson’s correlation and multicollinearity analysis, conducted with SPSS 25.0 software (SPSS Inc., Chicago, IL, USA). Highly correlated independent variables were excluded, and the tolerance limits and “dispersion expansion coefficient” (variance inflation factor; VIF) were examined. Correlation coefficient values greater than ±0.7 indicate high correlations, while very high correlations are represented by values greater than ±0.9. Multicollinearity was assumed to occur when the correlation coefficient was ±0.7 or greater. Tolerance limits of less than 0.1 and VIF greater than 10 also indicate multicollinearity between variables.

### 2.5. Quantification of Environmental Quality

The functional relationships between independent and dependent variables were tested through logistic regression analysis. Logistic regression analysis is often used when the dependent variable is binomial and independent variables are continuous, discrete, rank, or nominal. The logistic regression model is as follows:(1)$$\frac{pz}{1-pz}=a+b_{1}\chi_{1}+b_{2}\chi_{2}+\cdots +b_{p}\chi_{p}$$

Here, *PZ* indicates the probability that a reference category will occur in the dependent variable, a and *b* are unknown numbers, and *x* is a variable. From the regression coefficients estimated using this equation, we can obtain the following equation for the posterior probability:(2)$$p_{z}=\frac{\exp\left(a+b_{1}\chi_{1}+b_{2}\chi_{2}+\cdots +b_{p}\chi_{p}\right)}{1+\exp\left(a+b_{1}\chi_{1}+b_{2}\chi_{2}+\cdots +b_{p}\chi_{p}\right)}=\frac{1}{1+\exp\left(-\left(a+b_{1}\chi_{1}+b_{2}\chi_{2}+\cdots +b_{p}\chi_{p}\right)\right)}$$

Logistic regression analysis was performed with 14 variables that had been verified in the independence analysis. Variables with significance probabilities ≤ 0.05 were excluded. This process was repeated using Akaike’s information criterion (AIC), Nagelkerke R^2^, binomial deviance, and receiver operating curve (ROC) analyses until a suitable model was obtained. Then, correlation equations were derived, and a probability map of environmentally favorable areas was developed. In the logistic regression analysis, coefficients were derived for each variable, and the probability of favorable areas was estimated using the correlation equation. However, the weighted values of the variables suggested in the author’s previous study of the AHP technique could not be confirmed.

### 2.6. Development of the Environment Evaluation Map and Model Testing

A regression model was developed based on logistic regression analysis and tested through ROC analysis. A probability map of environmentally favorable areas was developed based on the regression model. The areas in the probability map were divided into “units” of 20%, with the areas with the highest protection values designated as grade 1 and the areas with the lowest protection values designated as grade 5. An environmental quality evaluation map was constructed using this 5-grade system. Railway projects are inherently linear in form. Therefore, environmental quality grades were averaged for each section and unit along the Gyeong-ui Central Line.

To examine the usability of the environmental quality evaluation map, kappa analysis was conducted based on the results obtained from the national land environment evaluation map. The similarity of the results was tested through a cross-sectional analysis of the descriptive statistics, from which the correspondence of the two maps was determined. Kappa values were calculated using the following equation:(3)$$k=\frac{\left(a-b\right)}{\left(1-b\right)}$$

*a*: Probability of correspondence between evaluators, *b*: Proportion of evaluations that are coincidentally correspondent between evaluators.

## 3. Results and Discussion

### 3.1. Independence Analysis

For independence analysis, multicollinearity and correlation analyses were conducted to examine the correlations between each pair of items. A strong correlation was obtained between the presence of legally protected areas and the number of adjacent protected areas (correlation coefficient = 0.994). The presence of legally protected areas was selected for further analysis, as it is a more important factor for evaluating the environmental quality of a site [3,4]. The correlation between the number of gaps and fragmented patch size was relatively strong, at 0.691, and the number of gaps was then excluded from the analysis. Multicollinearity analysis with the remaining 14 headings showed a VIF of less than 10 for all headings (Table 2). Therefore, all 14 headings were subjected to logistic regression analysis.

### 3.2. Numerical Assessment of Environmental Quality

Logistic regression analysis was performed using 14 headings with 274,689 samples. Headings with significance values > 0.05 were excluded. Logistic regression analysis was conducted repeatedly with the seven selected headings to identify the best model (Table 3). The conditions for determination were low AIC, low binomial deviance, and high McFadden R^2^. The model, including all seven headings, appeared to be the best model, as it had the lowest AIC (136,913.466) and binomial deviance (10,168.65) values and the highest McFadden R^2^ (0.377) value. The model’s sensitivity was 79.6%, and its specificity was 79.0%. The area under the curve (AUC) value obtained from the ROC test was 0.886.

In Table 3, B is the regression coefficient; a positive value indicates a higher probability of being a favorable area, while a negative value indicates a lower probability of being a favorable area. EXP (B) is an odds ratio; a positive effect is represented by a value exceeding one and a negative effect by a value below one. The results showed that headings such as ecosystem type, vegetation conservation class, tree age class, and fragmented patch size positively affected the probability of an area being favorable. Meanwhile, the headings of species diversity, ecological naturalness class, and presence of river ecosystem negatively affected this probability. The logistic regression model was constructed as follows:(4)$$\;\mathrm{Logit}\left(p\right)=5.926-2.031a_{1}+0.550a_{2}+0.043a_{3}+0.329a_{4}-1.672a_{5}-1.578a_{6}+0.007a_{7}\;$$

The logit obtained from Equation (4) can be substituted into the following equation.
(5)$$\mathrm{Probability}\;\mathrm{of}\;\mathrm{favorable}\;\mathrm{area}=\frac{\exp\left(p\right)}{1+\exp\left(p\right)}$$

ROC analysis of the final model showed an AUC value of 0.886. The fitted model showed 90.3% classification accuracy. Based on the logistic regression model obtained in this study, a probability map of environmentally favorable areas was created. In addition, an environmental quality evaluation map was constructed based on the classification of areas using the 5-grade system.

### 3.3. Environmental Quality Evaluation Map

The probability of environmentally favorable areas appeared to increase where vegetation was excellent, tree age was high, and patch area was relatively large. Otherwise, the probability decreased. The environmental quality evaluation map based on the 5-grade system had the following proportions: grade 1, 5.44%; grade 2, 19.56%; grade 3, 10.48%; grade 4, 21.55%; and grade 5, 42.97% (Figure 3). Nearly half of the area was designated grade 5, with low protection value, while 25% of the area was grade 1–2 (high protection value). Analysis based on the national land environment evaluation map issued by the government for evaluation of environmental quality showed differing proportions of grades 1 (22.82%), 2 (18.55%), 3 (1.99%), 4 (6.51%), and 5 (50.24%). This classification indicates that up to 40% of the railway line passes through areas with high protection value.

### 3.4. Validation of the Model

Kappa analysis was conducted for environmental quality evaluation, using the environmental quality evaluation map from this study and the national land environment evaluation map. Kappa analysis is a statistical measure that tests the correspondence of values in different categories (Cohen, 1968; [20]). For each measurement category, point values were taken from both the environmental quality evaluation map and the national land environment evaluation map. Kappa coefficients close to 0 indicate correspondence due to chance, while coefficients close to 1 indicate genuine correspondence.

Cohen’s kappa coefficient (0.201–0.4: a degree of correspondence, 0.401-0.6: a reasonable match, 0.601–0.8: a significant match, and 0.801–1.0: a perfect match) was 0.212, indicating that the two maps do not correspond well (Table 4). The national land environment evaluation map showed a grade 1 area of 56,376,785 m^2^. Logistic regression analysis showed 10,639,370 m^2^ of overlapping grade 1 area with the environmental quality evaluation map. Thus, the correspondence between the two maps was 18.87%. When the correspondence test was conducted for grades 1 and 2 combined, the overlap area was 54,799,415 m^2^ out of the total of 102,280,665 m^2^. The correspondence between the two maps was 53.58% in this case. In summary, the correspondence between the environmental quality evaluation map based on the results of this study and the national land environment evaluation map was relatively low. The low concordance between the results of this study and the national land environment evaluation map is the result of including the fragmented patch size item, which indicates the impact of the railway project, as the index of this study. As land use progresses, the impact of fragmentation must be considered.

## 4. Conclusions

The Gyeong-ui Central Line, which passes through both urban and rural areas, was used as the study site. Spatial information was obtained from each section through point sampling (the total number of sampling points was 274,689) and analyzed for independence. The number of adjacent protected areas and the number of gaps were excluded from the analysis due to high correlations with other factors. Logistic regression analysis was used to evaluate the environmental quality of the railway project area. Seven headings were selected: biodiversity class, ecosystem type, vegetation conservation class, tree age class, ecological naturalness, presence of river ecosystems, and fragmented patch size. Logistic regression analysis was conducted repeatedly and indicated that the model including all seven headings was the best model. Based on the regression model, a probability map of environmentally favorable areas and an environmental quality evaluation map were constructed. The classification accuracy was 90.3%, and the model fit was 88.6% based on ROC analysis. The Kappa coefficient was 0.212, showing that the correspondence between the environmental quality evaluation map from this study and the national land environment evaluation map was relatively low. The low concordance between the results of this study and the national land environment evaluation map is the result of including the fragmented patch size item, which indicates the impact of the railway project, as the index of this study. Especially, the model used in this study can compare the environmental ecology of the railroad development project with the national land environment evaluation map. As for the railway project, fragmented patch size, an item not included in other case studies, was added as an ecosystem cut-off project, and through this, the ecosystem cut-off and fragmentation were evaluated.

The results of this study can be applied to the “Guidelines for the Construction of Environment-friendly Railways”. Alternative railway routes could be determined based on our results for the Wolgot-Pangyo Line and Yeoju-Wonju Line, which are at the planning stage of strategic environmental impact assessment, thereby considering environmental value in railway project design. The environmental quality evaluation map can be used as basic data for determining priority areas for future railway projects. In all railway projects, examination of demands and economic feasibility is essential. Indicators related to ecological value can be added as factors for consideration in this decision-making process. The results of the present study could be applied to future railway projects to identify weaknesses in environmental quality. Furthermore, environmental quality problems related to railway projects may be mitigated through the application of such indicators. However, considering the topography of Korea, various types of railways such as mountain railways and underwater railways can be constructed. In the case of constructing a route that passes through forest areas and coasts, only grades 1–2 with high protection values may appear, and additional review will be required. Also, it is possible to consider the future development of the proposed model including other independent variables such as the socio-economic dimension of the environment-anthropogenic factors. In the future, like overseas technology trends, it is necessary to conduct a study on whether it is possible to construct various environmental and ecological time-series changes for the currently operating lines by using the UAV technology with the indicators and items developed in this study.

## Figures and Tables

**Figure 1 ijerph-18-11764-f001:**
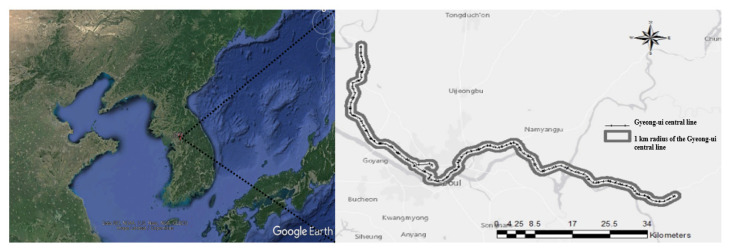
Gyeong-ui Central Line.

**Figure 2 ijerph-18-11764-f002:**
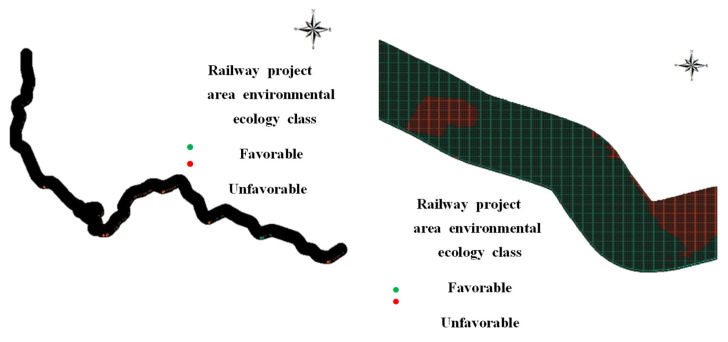
Evaluation of sampling points with raster data.

**Figure 3 ijerph-18-11764-f003:**
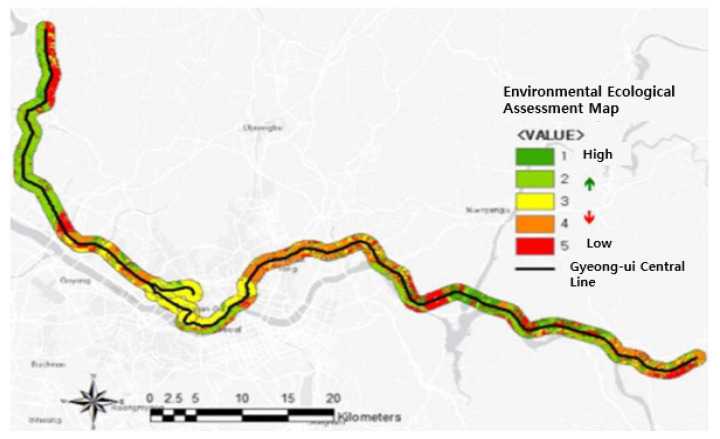
Environmental quality evaluation map of the Gyeong-ui Central Line.

**Table 1 ijerph-18-11764-t001:** Dependent and independent variables used for analysis of spatial information.

Variables		Values
Dependent variable (Railroad development areas environmental ecology conservation value area)		0: Absent
		1: Present
Independent variables	Species diversity	1: 1st grade
		2: 2nd grade
		3: 3rd grade and above
	Ecosystem diversity	-
	Species richness	1: Built and dry area
		2: Agricultural area
		3: Forest area
		4: Grassland
		5: Wetland
		6: Bare land
		7: Water area
	Vegetation conservation class	1: 1st grade
		2: 2nd grade
		3: 3rd grade
		4: 4th grade
		5: 5th grade
	Tree age class	1: 1st grade
		2: 2nd grade
		3: 3rd grade
		4: 4th grade
		5: 5th grade
		6: 6th grade
	Ecological naturalness class	1: 1st grade
		2: 2nd grade
		3: 3rd grade
	Presence of wetland ecosystem	0: Absent
		1: Present
	Presence of river ecosystem	0: Absent
		1: Present
	Fragmented patch size	-
	Rate of patch split	-
	Ratio of tunnels/ bridges/ecological passage	-
	Number of gaps	-
	Presence of legally protected area	0: Absent
		1: Present
	Number of adjacent protected areas	-
	Presence of the endangered species habitat	0: Absent
		1: Present
	Presence of endangered species	0: Absent
		1: Present

**Table 2 ijerph-18-11764-t002:** Results of multicollinearity analysis of 14 headings.

Heading	Non-Standardization Factor		Standardization Factor	T	Significance	Nominal Statistic	
	B	Standard Error	Beta			Tolerance	VIF
(Constant)	0.331	0.005		64.349	0.000		
Species diversity	−0.016	0.002	−0.005	−9.256	0.000	0.943	1.060
Species richness	−0.003	0.000	−0.014	−14.999	0.000	0.346	2.889
Ecosystem diversity	0.000	0.000	0.000	0.863	0.388	1.000	1.000
Vegetation conservation class	0.002	0.000	0.005	6.680	0.000	0.641	1.561
Tree age class	0.003	0.000	0.014	22.416	0.000	0.771	1.297
Ecological naturalness class	−0.095	0.001	−0.150	−179.45	0.000	0.458	2.185
Presence of wetland ecosystem	0.032	0.002	0.008	14.323	0.000	0.960	1.042
Presence of river ecosystem	−0.041	0.001	−0.033	−39.630	0.000	0.452	2.212
Fragmented patch size	−2.15 × 10^−5^	0.000	−0.006	−8.918	0.000	0.695	1.440
Rate of patch split	−2.15 × 10^−5^	0.000	−0.003	−5.347	0.000	0.811	1.234
Ratio of tunnels/ bridges/ecological passage	0.013	0.002	0.003	5.415	0.000	0.999	1.001
Presence of legally protected area	0.920	0.001	0.882	1346.489	0.000	0.746	1.341
Presence of endangered species habitat	−0.002	0.006	0.000	−0.315	0.753	0.982	1.018
Presence of endangered species	−0.002	0.051	0.000	−0.041	0.967	0.987	1.013

**Table 3 ijerph-18-11764-t003:** Results of logistic regression analysis of seven headings.

Variable	B	Standard Error	Wald	Significance	EXP (B)	95% Confidence Interval for EXP(B)	
						Lower Limit	Upper Limit
Species diversity	−2.031	0.041	2410.020	0.000	0.131	0.121	0.142
Ecosystem diversity	0.550	0.005	10,278.020	0.000	1.734	1.715	1.752
Vegetation conservation class	0.043	0.007	39,325	0.000	1.043	1.030	1.057
Tree age class	0.329	0.004	6790.892	0.000	1.390	1.379	1.401
Ecological naturalness class	−1.672	0.015	12,531.986	0.000	0.188	0.182	0.193
Presence of river ecosystem	−1.578	0.030	2756.880	0.000	0.206	0.195	0.219
Fragmented patch size	0.007	0.000	5229.324	0.000	1.007	1.007	1.007
(Number)	5.926	0.120	2435.163	0.000	374.664		

**Table 4 ijerph-18-11764-t004:** Results of kappa analysis based on values from the national land environment evaluation map and environmental quality evaluation map.

(Unit: m^2^)							
		National Land Environment Evaluation Map					
		Grade 1	Grade 2	Grade 3	Grade 4	Grade 5	Total
Environmental quality evaluation map	Grade 1	10,639,370	1,985,884	422,911	179,062	207,856	13,435,084
	Grade 2	30,417,225	11,756,936	1,505,385	1,371,313	3,302,308	48,353,166
	Grade 3	3,536,259	10,121,079	1,082,473	2,683,238	8,483,422	25,906,471
	Grade 4	4,804,993	9,340,043	546,186	4,964,260	33,612,456	53,267,936
	Grade 5	6,978,937	12,699,938	1,086,072	6,886,257	78,554,437	106,205,642
	Total	56,376,785	45,903,880	4,643,027	16,084,129	124,160,479	247,168,300

## Data Availability

Not applicable.

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
