# Peer review of "Quantification of the Ecological Value of Railroad Development Areas Using Logistic Regression Analysis"

_ijerph, 2021, doi:10.3390/ijerph182211764_

Round 1

Reviewer 1 Report

The manuscript is, in general, well written and organized, but there are several changes or improvements to make, regarding the content and the format of the document. Therefore, in order to improve the quality of your article please include the following changes/adjustments/improvements:

  • the "Literature review" part should be better delimited, possibly included as a separate section;
  • in some places the studies in the literature review are presented too succinctly, some of their results requiring more detail;
  • the introduction can end with a brief presentation of the structure/sections of the article;
  • in table 1, in which the presentation of the variables used in the model was made, a suggestive name for the dependent variable should be established;
  • it is not shown whether the logistics model used in this article has also made an improved selection of independent indicators / variables, compared to those used in other similar models, or what other improvements are made by this model compared to other existing models. It is only shown that indicators and headings proposed by Kim et al. (2019) were used. A future development of the proposed model can be considered by including other independent variables (for example, indicators for measuring the socio-economic dimension of the environment - anthropogenic factors);
  • it should be specified more clearly whether the proposed model is the first of its kind on the basis of which maps have been built or there are other similar models;
  • the way of defining the classes for the different independent variables should be specified (Ex. vegetation conservation class - grades 1,2,3,4), and if these classes existed and were already used or they were created by the author;
  • not all the symbols that appear in relation (4) are explained (a1, a2, ..., a7);
  • should be explained the socio-economic considerations and consequences of this "low correspondence” between the environmental quality evaluation map from this study and the national land environment evaluation map;
  • in presenting the research results, references should be made to the results of other studies (signaling similarities / discrepancies with other studies);
  • any limitations of the study should be specified;
  • future research directions of the analysis performed in this study should be presented.

Reviewer 2 Report

Authors must pay attention to the requirements of the journal. The conclusions only partially show the results made by the authors.
It should be detailed.

Reviewer 3 Report

Dear Authors

The issue raised in the article is quite interesting. Unfortunately, the article requires significant corrections and additions. The content of this manuscript does not meet the standards adopted for scientific articles. The text submitted for the review can only be used as a basis for a future article.

My main comments and suggestions are as follow:

  1. IJERPH (International Journal of Environmental Research and Public Health) is an international journal, that is why the authors should take into account the state of INTERNATIONAL knowledge.

The authors used only 18 bibliographic sources (all of national origin). Apart from their small number, I also have concerns about their quality - there are no significant international literature items there:

  • Items 1, 2 and 5 - are government / ministerial documents
  • Items 3, 9, 10, 12 and 13 - are doctoral dissertations
  • Items 4, 6, 7, 8, 11, 14, 15, 16, 17, and 18 are (national) articles published in Korea.
  • 6 items out of 18 (33.3% of the total number) are authors' own publications, so many assumptions comes from their previous research. What about the other researchers?

It is necessary to strengthen the article with other international publications.

  1. The main results of the research are missing from the ABSTRACT.
  2. The introduction should better describe a problem of research and show the main aim of the paper. INTRODUCTION lacks hypotheses / research questions. As a result, the reader does not know for what purpose the research is carried out, what they are supposed to give, what problem they can or should solve.
  3. An additional part should be separated from INTRODUCTION - LITERAUTRE REVIEW. It is necessary to indicate, for example, the "Ecological Value of Railroad" appearing in the title, also in terms of already completed research (including international research).

The Authors have to indicate the state of the art in this area, as well as the contribution of the manuscript to the theory (LITERATURE GAP is missing). Without that, I can seriously doubt whether the article will have any value for the international research community. At the moment, the article looks like a guide for the government.

Many sentences are not reflected in the literature sources or in the research carried out by the authors, eg.:

  • „Investment in railways has increased due to the need for environmentally friendly transportation”

How do the Authors know that? The literature sources is needed.

  1. Discussion/imitations are missing in the article.
  2. CONCLUSION is very short, general and fuzzy. The main conclusions can be listed.

Round 2

Reviewer 1 Report

The article can be accepted for publication in the present form.

Reviewer 2 Report

Thank you for making the corrections by noting my comments.
The article can be published.